# Algorithm for Process Innovation by Increasing Ideality

**Vladimir Sojka * and Petr Lepsik**

Faculty of Mechanical Engineering, Technical University of Liberec, 461 17 Liberec, Czech Republic; petr.lepsik@tul.cz
* Correspondence: vladimir.sojka@tul.cz

**Abstract:** Continual efforts to have better processes lead us to search for new ways to improve and innovate. One of the most powerful approaches to innovating technical systems is the TRIZ (Theory of Inventive Problem Solving). TRIZ is, unfortunately, very hard to learn and adequately use. This paper introduces a new comprehensive algorithm for the innovation of processes in production based on TRIZ principles. The Algorithm for Process Innovation by Increasing Ideality (AP3I) helps search for innovative ways to improve or change the process—either the whole process or its segments. Besides the original TRIZ, AP3I is easier to use and might be applied by engineers in industrial practice. On the other hand, results from AP3I are probably weaker compared to full TRIZ. Still, the AP3I can be very helpful in efforts to improve processes and can provide powerful ideas. The overall algorithm is also demonstrated in case studies on processes of packing and assembly.

**Keywords:** process innovation; process improvement; TRIZ; packing process; assembly line

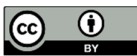

## 1. Introduction

There is a general need for continuous improvement in production processes. Innovative solutions to change processes must be found to achieve radical improvements. therefore, companies should focus on the innovation of their production processes. However, a problem can arise in choosing a method or approach to effectively generate innovative solutions.

There are many ways to approach the improvement in processes. Currently, the most commonly used are Lean Production and Six Sigma. Lean Production focuses mainly on reducing activities that do not create value. The overall goal of Lean Production is to have the best quality, lowest cost, shortest lead time, best safety, and high morale [1]. Six Sigma is focused mainly on the quality of the production. It is a data-driven method that reduces the variability in the process outcomes, leading a reduction in the number of defects. The goal of Six Sigma is zero defects, or, as the name of the method says, only 3.4 defects per one million opportunities are allowed, which is a six sigma probability in the normal distribution [2–4]. There are also other tools and methods for process improvement. The Theory of Constraints (TOC) focuses on throughput and aims to improve the weakest step of the process.

To mention other tools used for process improvement, the Design of Experiment (DOE) can search for the optimal setting of process parameters [5]. FMEA, or FMECA (Failure Mode and Effects Analysis/ Failure Mode Effects and Criticality Analysis), identifies possible mistakes and defines its counteraction and preventive action [6–8]. Quality Function Deployment (QFD) uses customers' needs to better define products or processes [9]. Predetermined motion time systems such as MTM (Methods-Time Measurement) or MOST (Maynard Operation Sequence Technique) improve time efficiency and the ergonomics of manual production activities [10,11]. Computer process simulation can help identify and resolve some hidden process problems and improve costs and production time of processes [12,13].

The currently used approaches for process improvement have many powerful aspects. However, when it is time to generate solutions to identified problems, tools such as brainstorming, trial and error, or guessing are often used to find a way to improve or innovate.

Brainstorming can be helpful when an already known type of problem is solved. Brainstorming does not lead to an ideal solution when a more complex problem occurs [14,15].

A more powerful tool or method for searching for ideas on how the process could be innovated is therefore needed to achieve radical improvements and innovation. With innovative ideas, production processes can be improved much faster. One of the most powerful methodologies for innovation is TRIZ (Theory of Inventive Problem Solving) [14,16,17].

This paper aims to develop a new algorithm for the innovation of production processes. The algorithm is based on TRIZ principles. It presents practical case studies on the generation of an innovative solution to the already improved packing process and an assembly line for heat exchanger units.

Lean Production or Lean Manufacturing is a method based on the Toyota Production System (TPS) [18,19]. The main focus is continuous improvement and respect for people. The Lean system is often represented as a house with two main pillars—Just-in-time (JIT), and Jidoka (also be described as autonomy). Many tools and methods of Lean support the house [20]. The goals of the Lean system are inside the roof of the house—best quality, lowest costs, shortest lead time, best safety, and high morale. These are the goals of Lean [1]. To achieve these goals, companies constantly improve their processes by eliminating waste. There are three types of activities that use Lean—value-adding activities, non-value-adding activities and non-value-adding but necessary activities [21]. Lean Thinking is the philosophy that may lead to successfully applying Lean to the company. It contains five principles—value, value stream, flow, pull, and perfection [22–24].

Six Sigma is a data-driven method for the improvement in processes. The name Six Sigma comes from its general goal: to reduce the variability in process outcome to the level of ±6 sigma of normal distribution. The result is the reduction in defects to a maximum of 3.4 defects per one million opportunities. A DMAIC (define, measure, analyze, improve, control) improvement cycle is used to apply the Six Sigma method. [2–4,25–27]. A combination of Lean and Six Sigma is often used and is referred to as Lean/Six Sigma (LSS). It uses benefits from both methods—a broad philosophical base from Lean and the data-driven approach of Six Sigma. [28–31]

An innovative approach should be used to achieve more significant improvements. The review results on open innovation [32] showed the need for process innovation. Zhu [33] shows that a lack of process innovation is also connected to slower labor productivity growth. Additionally, the importance of green- or eco-innovation has been declared by many authors [34–37].

TRIZ is a theory based on the research of millions of patents—in that research, repetitive patterns were found [38]. The same types of problems repetitively appear, to which generally the same kind of solutions can be found. Based on these findings, tools and methods of TRIZ theory were designed. These tools and methods help to overcome problems with highly innovative results. The solutions given by TRIZ follow the idea of the ideality and evolution of technical systems; they are not affected by psychological inertia. [39–43]

The literature review below, on the goals of using TRIZ with process improvement tools or TRIZ itself to improve processes, has several results. Many authors have tried to integrate TRIZ with different tools, but the results are too abstract or complex for practical use. TRIZ is hard to learn, so TRIZ is still not in the toolset of industrial engineers in practice. The review's outcome is that there is no need for new attempts to integrate TRIZ with other tools or methods. Instead of trying to combine TRIZ with other methods, a whole new method should be designed. A new method is suggested that is based on TRIZ principles but relatively easy to use by engineers in practice [44].

## 2. Methods

To find a way how to innovate production processes effectively, firstly, a literature review was conducted. The use of TRIZ tools and principles for process improvement and TRIZ integrated with other methods for process improvement is described above.

The main idea is that when the process's ideality is increased, the process is improved in a good way, i.e., closer to the ideal final result. The idea that changes the process to rapidly increase ideality should be highly innovative.

Based on this premise, a method for resolving problems by increasing the ideality (3I—Innovation by Increasing Ideality) was designed. The next step was a design of an algorithm for systematic process innovation based on the 3I problem-solving method. The final step was a practical experiment with applications in real processes, where the process was firstly improved by the classical approach and then by the newly proposed algorithm.

The idea is that when a process is repetitively improved by classical tools, the yield of improvement decreases with each improvement project. We presume that the yield decreases more and more until the process meets its optimal state. When the process is innovated, a theoretical optimum level is changed as well. Based on this assumption, the real process should be improved by the classical approach, and the newly proposed method should be applied to the improved state. Then, the yields of improvement should be compared. If the yield of the second improvement is bigger than the first (classical) improvement, our proposed method provides innovative solutions for radical process improvements.

*2.1. Ideality*

As a primary measure unit, ideality was used. TRIZ ideality is described by Equation (1) below,

$$I = \frac{\sum_{i=1}^{n} B_i}{\sum_{j=1}^{m} H_j} \tag{1}$$

where *I* is the degree of ideality; *B* is the benefits—all positive functions' effects; *H* is the harms—all negative functions' effects; and *n* and *m* are a number of beneficial harmful effects, respectively. Some authors also use costs as a separate aspect of harm. This kind of equation is too abstract, and it would be complicated to calculate the degree of the ideality of process states [45–48].

Based on the above, a new equation for the calculation of the ideality degree of the processes was developed. A new equation is based on the extended TRIZ ideality equation that includes costs. For the process ideality, every part of the equation is clearly described. A general equation for process ideality is then (2):

$$Ip = \frac{\sum_{i=1}^{n} PO_i}{\sum_{j=1}^{m} NI_j + \sum_{k=1}^{o} NO_k} \tag{2}$$

where *Ip* is the ideality of the process, *PO* is the positive outcomes of the process, *NI* is the needed inputs for the process, and *NO* is the negative outcomes of the process. *n*, *m*, and *o*, are the numbers of positive outcomes, needed inputs, and negative outcomes. The equation can also be expanded as shown below (3):

$$Ip = \frac{\sum_{i=1}^{n} POd_i + \sum_{r=1}^{p} POe_r}{\sum_{j=1}^{m} NI_j + \sum_{k=1}^{o} NO_k} \tag{3}$$

where positive outcomes are divided into *POd*, which represents the demanded positive outcomes, and *Poe*, which represents the extra positive outcomes that are above expectations, with *r* extra positive outcomes. For better understanding, every parameter of the equation is described further.

*POd*—Demanded positive outcomes (variation in TRIZ's Benefits) represents the final outcome (product or service) for which the process exists. For example, it can be a manufactured, assembled, or transported part. From a service point of view, the outcome can be a provided action, such as sold units, transported people, or finished haircuts. *POd* is represented as 1 in the equation because the equation is related to the production of one demanded unit of (*PO*).

*NI*—Needed inputs (variation in TRIZ's Costs) for the process. This can be understood as all income needed to produce demanded unit (*PO*). There can be "costs" in the case of materials, energies, people, equipment, financial costs, and time.

*NO*—Negative outcomes (variation in TRIZ's Harms) are all other outcomes of the process that are not demanded. There can be waste materials, NOK parts, and negative effects on the environment or health.

*POe*—Extra positive outcomes, or positive outcomes above expectations, are outcomes that do not fulfill the demanded outcome, but they have some other positive effect on the process or its surroundings. This can be, for example, when a process as a side outcome produces electric energy, consumes $CO_2$, etc.

Costs and harms can be divided into parts representing particular costs (incomes) or harms (negative outcomes). An example of a full equation can be seen below in Equation (4).

$$Ip = \frac{POd + POe}{\left(T + F + P + Eq + E + M\right) + \left(W + NOK + Sp + Se + O\right)} \tag{4}$$

where *POd* is the unit of demanded outcome (1 pcs); *POe* is the extra positive outcomes related to *POd*; *T* is the time needed to produce the demanded outcome unit; *F* is the financial costs required for the production of demanded outcome unit; *P* is the number of workers in the process; *Eq* is the equipment needed (evaluation of the number and complexity of the necessary equipment to produce demanded outcome unit); *E* is the energy required to produce demanded outcome unit; *M* is the materials needed to produce demanded outcome unit; *W* is the wastes from the process (related to the unit of demanded outcome); *NOK* is the number of NOK parts related to the unit of demanded outcome; *Sp* represents safety for people (safety, health, and ergonomics), which means a level of threat to persons related to the unit of demanded outcome; *Se* represents the environment's safety (surroundings), which is a level of threat to the environment and surroundings related to the unit of demanded outcome; and *O* is for other harmful outcomes, in case something was not covered.

The original state of the process is set as the initial state (with the sign of 0). For the original state, all parameters are set to one (actual values must also be collected). After improvement, values of process parameters are compared to original ones, and values related to ones from the original state are put into the equation. As an example with the time parameter, the original state has a production time of 15 min, and *T* in equation ($Ip_0$) will be 1. After improvement of the process, throughput is 13 min, and in equation ($Ip_1$), the T will be 0.87 (improved 13 divided by original 15). Overall ideality is affected positively, and it does not matter if there are more or fewer value-adding activities.

### 2.2. Innovation by Increasing Ideality

As a core of the Innovation algorithm, an Innovation by Increasing Ideality (3I) was developed. This method contains several steps that lead the solver through the problem-solving process. The 3I method can be used to overcome process-related problems by innovating the step, or segment, where the problem occurs.

The steps of the Innovation by Increasing Ideality method are:

- Problem;
- Purpose;
- Principle;
- Ideal state;
- Ideality questions;
- Inspiration in effects and trends;
- Contradiction;
- Other TRIZ tools;
- List of solutions;
- Final solution.

The first step is to define and describe the problem—what the problem is and when and where it occurs. The next step is to describe the purpose of the process where the problem occurs—i.e., the goal of the process. The principle of the process is based on what principle is the currently achieved purpose. In the ideal state step, the ideal state of the process is determined (for the description, words such as "itself" should be used). Ideality questions should be made to approach the ideal state (to remove the gap between the current and the ideal state). Inspiration refers to looking into scientific effects, trends, and databases for an existing solution or idea. Contradiction refers to determining and resolving contradictions (technical and physical contradictions). Other TRIZ tools can be used if needed—this step depends on the solver knowledge of TRIZ; for example, the sDTC operator, method of little men, and other TRIZ tools can be used to generate innovative ideas and avoid psychological inertia. List of solutions and ideas refers to all ideas and solutions found in previous steps being collected together in a clear list. For the final solution, based on a list of solutions, a final solution is found (another round of inspiration in databases for the existing solutions can be helpful).

The 3I method was introduced in detail in a study [49], where a problem with the transportation of heat exchangers was innovatively resolved.

*2.3. Development of the Algorithm for Process Innovation by Increasing Ideality*

The algorithm for Process Innovation by Increasing Ideality (AP3I) is based on the method of Innovation by Increasing Ideality (3I) described above. The main idea is that the process is systematically divided into segments. In these process segments, a 3I method is applied. There is no need for existing problems in the segment. The 3I method can be used just to search for ideas on how to innovate the process or its segments. If there is no acceptable solution for innovation, the segments are repetitively divided into even smaller segments and finally into process steps. The 3I method is used for all segments or steps where good solutions were not found in the segment of higher level. The whole algorithm is visualized below in the Figure 1.

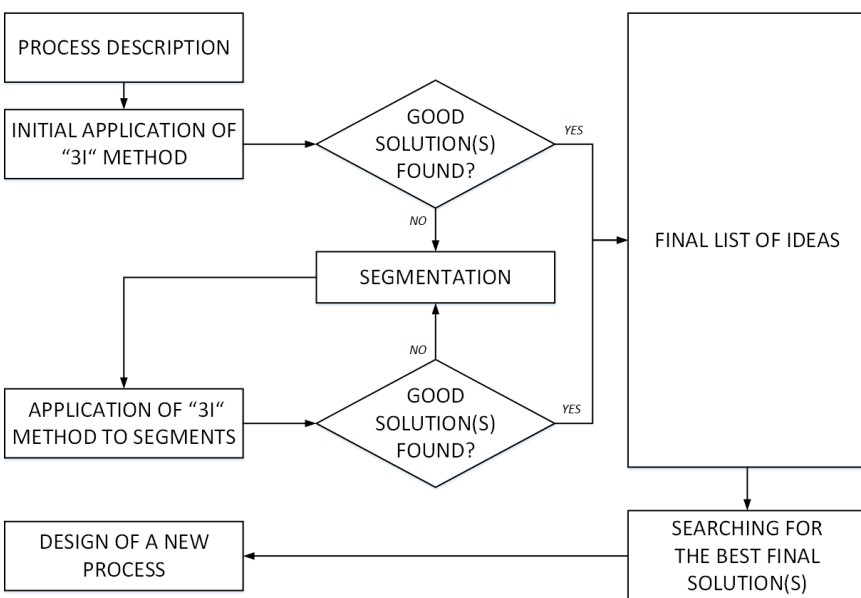

**Figure 1.** A scheme of the Algorithm for Process Innovation by Increasing Ideality.

Firstly, a process on which the application of AP3I aims is described. The definition of borders of the process is crucial. Additionally, it is good to know the inputs and outcomes of the process. Tools such as SIPOC (Supplier, Input, Process, Output, Customer) and process mapping can be used.

The Innovation by Increasing Ideality method is used for the whole process. The whole process's purpose, principle, and ideal state are described. After that, ideas for innovation of the whole process are generated using ideality questions, inspiration from scientific effects and databases, and determination and overcoming of contradictions. From the list of solutions, the final solution for the innovation of the whole process can be found—an entirely new concept of achieving the purpose of the process. If an idea powerful enough for innovation of the process is found, the algorithm could end, and efforts could continue to design the new process based on the idea found by AP3I.

A good solution or idea for innovating the whole process cannot always be easily found. In the case that it is not found, the solver is led to the next phase of the algorithm. The process is divided into segments—sub-processes, for example. On each segment, the 3I method is applied. New ideas on how to innovate the process segments are collected together. Again, if powerful ideas for innovating the process (all segments) are found, the solver can go into the designing phase. If not, the process is divided deeper into smaller segments, and the application of the 3I method for each new segment continues. The segmentation of the process ends on the level of process steps or activities.

After the 3I method is applied to all remaining steps, ideas are collected in the final list of solutions. Even if no powerful, innovative solution to some segment is found, some ideas can appear—these ideas are also collected into the final list of solutions. Similarly, when the solver has already found a good solution to the innovation of the process segment, the algorithm can continue on lower sub-segments of the already resolved segment. This can lead to interesting solutions to operations in the process, which can be used on a different process or new process design.

The best final solution, or solutions, should be quite easily found from ideas in the final list of ideas. In the next step, a new process is designed based on the final solution. After a new process based on the found solutions is designed, the planning and realization phase can begin. Findings, ideas, and solutions can also be used to improve other similar processes in production.

### 2.4. Case Study—Packing Process

Application of the algorithm for process Innovation by Increasing Ideality was firstly used on the process of packing parts from glass and metal. As a measurement of the algorithm's functionality, a comparison of the changes in the degree of process ideality and other indicators was used. Firstly, the process was improved using the classical approach, and on the improved process, the AP3I method was applied. Process ideality, defined by Equation (4), was used to measure the improvement rate.

An improved process is a process of packing parts and sub-assemblies from glass and metal in the Preciosa-Lighting company in the Czech Republic, where luxury custom chandeliers are made. Because of custom production, there is a high variability in part types and dimensions, so there are many packaging types and many packing materials.

The first improvement in the packing process was based on a Lean toolset. A DMAIC (Define, Measure, Analyze, Improve, Control) cycle drove the project. The process was mapped, and a process map was created. After that, visual observation of the process steps was conducted together with a workers' time sampling of the workday. From the results, a value analysis was made. Moreover, several measurements on how to reduce non-value-added activities and times in the packing process were made. The main result was a re-layout of the whole packing hall. To design a new layout, a few more tools were used: the measurement of the intensity of flows, a spaghetti diagram, a storage reduction, and the frequency of using different types of transport units.

Original and improved layouts with the material flow can be seen in the Figure 2 below.

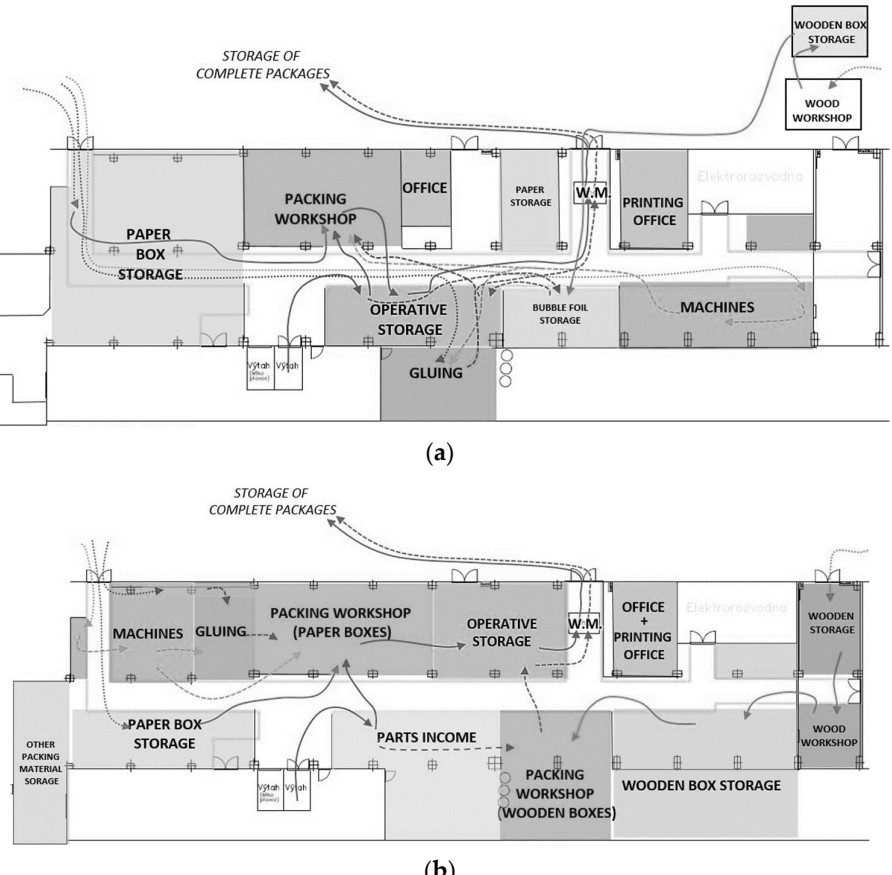

**Figure 2.** Material flows of packing process: (**a**) original state; (**b**) improved state [50].

With the re-layout, several minor improvements to the process were also made. New packing tables and packing benches and more weighing machines were installed. The overall improvement was measured by the length of parts throughput, which was shortened by 47.7%. The improvement results are summarized in the Table 1 below.

**Table 1.** Comparison of the original and improved state of the packing process.

| Parameter | | Original State | Improved State |
|---|---|---|---|
| Length of parts' throughput | (m) | 82.6 | 47.7 |
| Sum of workers' walking distance | (m) | 78.3 | 59.5 |

The Algorithm for Process Innovation by Increasing Ideality was applied on the improved packing process. Firstly, the 3I method was used for the whole packing process. After that, the process was divided twice into smaller and smaller segments. The 3I method was applied to each segment. The application to the whole process is summed in a Table 2 below.

**Table 2.** An application of the 3I method to the whole packing process.

| 3I Step | Content |
|---|---|
| Problem/Process | Packing process of glass and metal parts |
| Purpose | To pack parts into transport packaging. The package should protect parts during transportation to the final destination. |
| Principle | Parts are put together with soft materials into boxes (paper, wooden), boxes are transported to the customer. |
| Ideal state | - Parts are packed themselves. <br> - Parts are protected by themselves. <br> - Parts are safely transported to the final destination. |
| Ideality questions | - How to change the process to achieve easier packing? <br> - How to protect parts without packing? <br> - How to safely transport parts without packing? |
| Trends and effects | Segmentation, Dynamization, <br> Protect-solid; move-solid; |
| Technical contradiction(s) | TC1: 39. Productivity should be improved, /36. Device complexity should not worsen. <br> 12. Equipotentiality; 17. Another dimension; 28. Mechanics substitution; 24. Intermediary <br> TC2: 27. Reliability/39. Productivity <br> 1. Segmentation; 35. Parameter changes; 29. Pneumatic and hydraulics; 38. Strong oxidants <br> TC3: 9. Speed/27. Reliability <br> 11. Beforehand cushioning; 35. Parameter changes; 27. cheap and short-living objects; 28. Mechanics substitution |
| Physical contradiction(s) | None |
| Other tools | None |
| List of ideas | Automation, aerogel, auxetics materials, bubbles, foam, origami, lamination, use of universal air-bags <br> Equipotentiality, Another dimension, Mechanics substitution, Intermediary, Segmentation, Parameter changes, Pneumatic and hydraulics, Strong oxidants, Beforehand cushioning, Parameter changes, Cheap and short-living objects, Mechanics substitution |
| Solution | Not satisfying |

After applying the 3I, the ideas that were generated for where to search for innovative solutions were collected in the list of ideas. No good solution was found, so the process was divided into three segments—box preparation, packing, and package processing. The 3I method was applied to all three of these segments.

We used packing material as a box, stored packing materials at the packing place, and used gravitation or static electricity. These are ideas generated from using the 3I

method for the three segments of the packing process. Because there is still no satisfying solution, the process was segmented deeper.

The smallest segments are process steps. These steps are walking for parts, bringing parts to the workplace, folding the box, taping the box, packing, documentation, closing the box, bringing the package to the strapping machine, strapping, writing on the box, bringing the box to the weighing machine, weighing the box, waiting for the printing of the information, sticking information on the box, bringing the package to a palette, and fixing it together. All steps with no change on the product (transport, walking, or waiting) should be eliminated.

After the 3I method was applied to process steps, ideas and solutions were put together into the final list of ideas. From the list, solutions for problems in the process could be found, as well as ideas to make something in the process better. The Table 3 below summarizes ideas and possible solutions.

**Table 3.** Comparison of the original and improved state of the packing process.

| Idea/Possible Solution | Resolved Problem/Improved Step |
|---|---|
| Use of light-sensitive areas on boxes | Printing by light |
| Packing on weighing-machine | Weighing—during processing |
| Weighing—machine in a transport unit | Weighing—during processing |
| Electronic documentation | Elimination of printing of documentation; Elimination of putting documentation into the box |
| Sticky materials | Gluing of boxes together/into the shape |
| Magnets | Closing of boxes |
| Gravitation and cylinders (conveyors) | Automatic folding (building) of boxes during the transport |
| Gravitation | Putting parts into boxes |
| Static electricity | Packing material around the parts |
| Package out of packing material | Elimination of boxes |
| Transport by gravitation conveyors | Optimization of material transport |
| New material or structure (aerogel, auxetic material, bubbles, lamination, foam, origami, …) | Innovation of packing material |
| Transport in air-bags | Instead of boxes and packing material, use bags filled by air (repetitive use) |
| Paper palettes | Stability during transportation of packages |
| Membranes inside a firm frame | Instead of packing material. Use of membranes in frames (universal) |
| Mushrooms as packing material | Ecology |
| Addable wheels to the boxes | Easier transport |
| "Paper wrap" | Paper instead of bubble foil |
| Use of tools and materials needed for final assembly as packaging or packing materials. | Reduction of packing material |

Based on the analysis of the ideas and possible solutions, a raw design of a new process was proposed. The main idea was to combine ideas to obtain the best process state. Additionally, inventive principles and ideas that were found from each step were considered to search for the final solution. The new packing process proposal was as follows:

1. Gravitation conveyors transport parts to the packing station (in plastic boxes, which are returned by a secondary gravitation conveyor back);

2. Parts are packed into the frames with soft and flexible membranes. Frames are then put into foldable boxes from wood or plastic (all frames and boxes have a universal dimension and are stored close to the packing station);
3. Filled frames are put into the foldable box. The box is on the weighing machine during the filling;
4. After the packing is complete, the weight of the package is read (because only universal parts are used, the known weight of used frames and the box can be easily calculated to find the net weight of the packed parts);
5. Weight and other information are written in small form and put into the small pocket on the box (this step can be further automatized by a printer connected to the weighing machine);
6. Gravitation conveyors transport the complete box to the preparation station for the final export;
7. The box has holes so that wheels can be attached to the box. Several boxes can also be easily connected.

Documentation is sent by electronic services, which leads to a reduction in paper and ink usage. Boxes are repeatably usable and foldable so that they are much smaller during the storing or transporting back to the company. Boxes can also be used as a transport unit for tools and materials needed for final installation (holders inside the box). The Figure 3 below shows a schematic of the membrane–frame system.

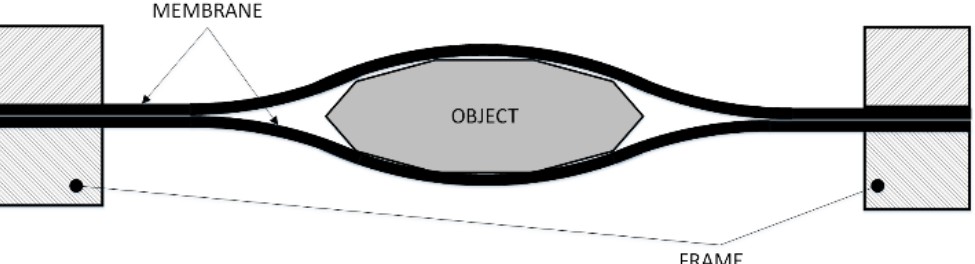

**Figure 3.** Scheme of new packing approach.

For the design of boxes and frames, it is recommended that its own variants are designed with the correct dimensions and all needed functions. As an inspiration, already existing boxes by Mireia Gordi Vila (in a work called "Fragile" by Mireia Gordi Vila, RCA 2014—mixed media) can be seen in the Figure 4 below.

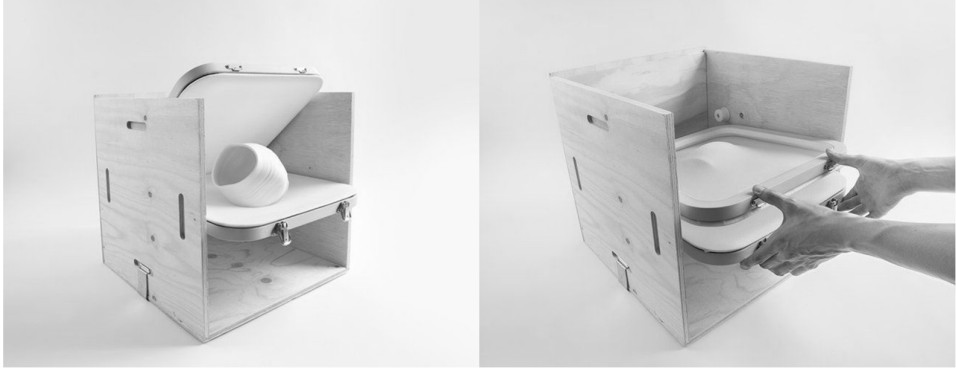

**Figure 4.** Existing system of frame-membrane packing "Fragile" by Mireia Gordi Vila. [51–53].

By realizing the new process proposal, all packing materials are eliminated to re-usable frames and foldable boxes. There are few frame types, so most of the storage of paper boxes can be reduced. Workers do not need to walk through the packing hall for boxes and materials. Several steps are combined, as weighing and packing occur at the same time.

### 2.5. Case Study—Process of Heat Exchanger Units Assembly

In the second case study, the algorithm is applied to the process of the assembly of heat exchanger units in the FläktGroup Czech Republic company (Liberec, Czech Republic). The heat exchanger is assembled together with metal sheet parts at the assembly line. Most of the joints are made by riveting. The assembly line also consists of testing and packing of the units. The Figure 5 below shows the assembly line and its layout. The real assembly line is shown in the Figure 6.

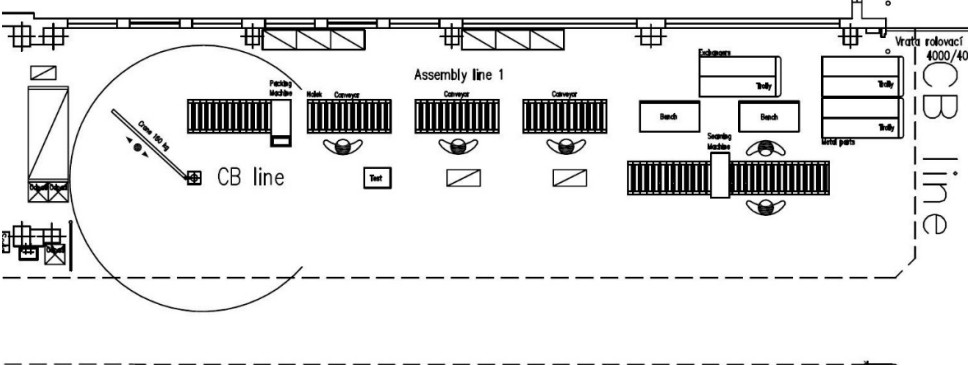

**Figure 5.** Layout of the assembly line.

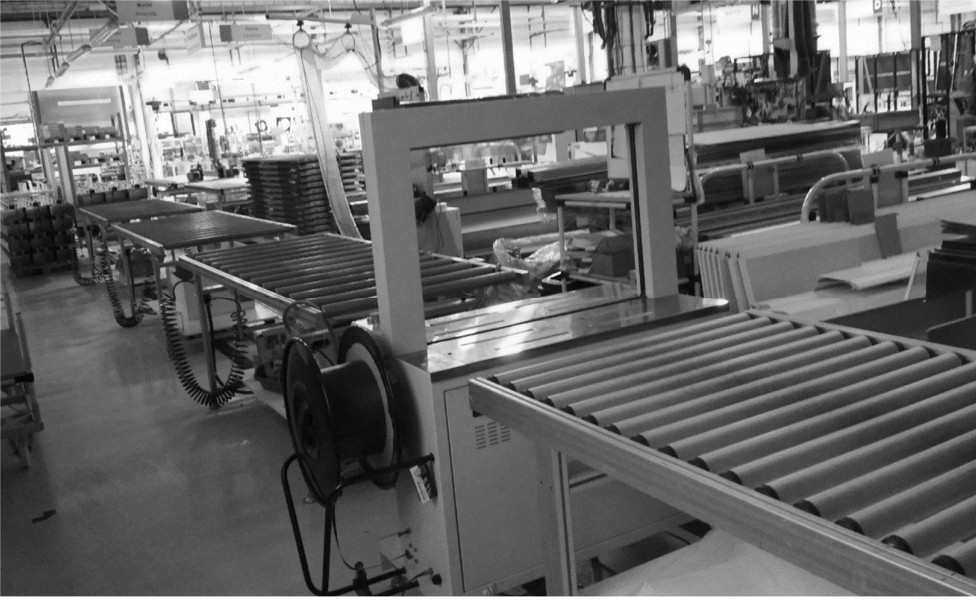

**Figure 6.** Assembly line.

A computer simulation was chosen as a method for the first improvement in the process. The simulation of the production process provides a good description of the system, and can quickly make different versions and compare them with each other. Siemens Tecnomatix Plant Simulation 14 software was used to create the process model and its analysis. Firstly, a model of the original process state was created. See the model in the Figure 7. The simulation time was set to 10 days.

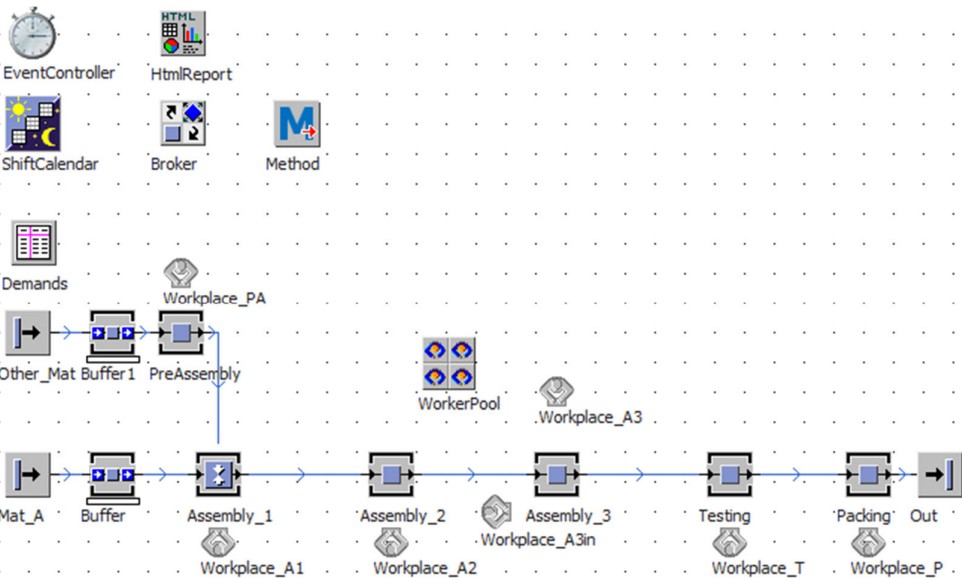

**Figure 7.** Simulation model of the assembly line—original state.

Based on the simulation results, several changes to the process were proposed. In a proposed improvement, the manual transportation of parts was eliminated. Additionally, an improvement in parts throughput in the process bottlenecks was made. The smooth flow was achieved by adding buffers with a capacity of one part between steps. The simulation model of the proposed improvement can be seen in the Figure 8 below. The possible amount of produced parts was improved by 39.6%. The original process could produce 237 parts in 10 days. After the proposed improvement based on the simulation, the process could produce 331 parts in 10 days.

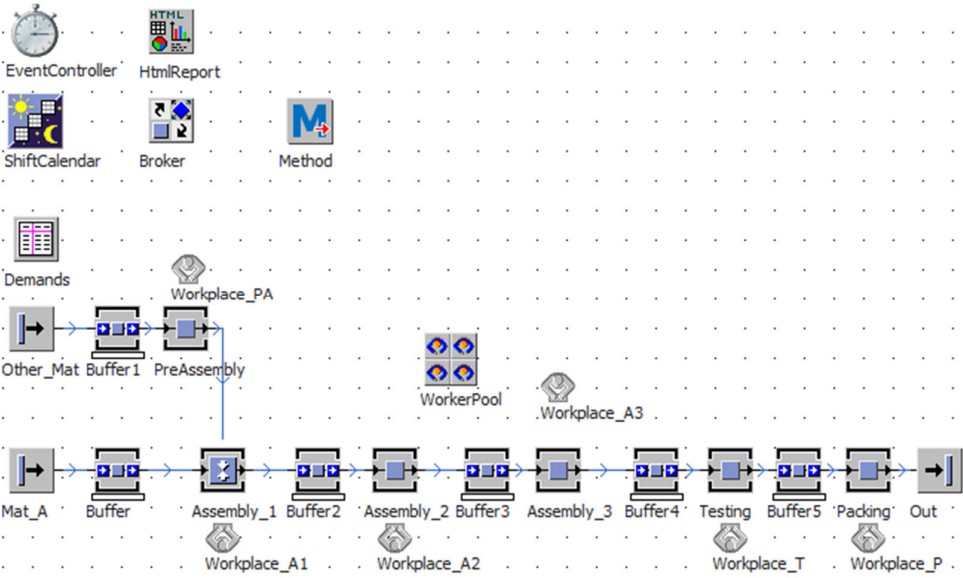

**Figure 8.** Simulation model of the assembly line—improved state.

The algorithm for Process Innovation by Increasing Ideality was then applied to the improved state of the process. Firstly, ideas for the innovation of the whole process were collected by using the 3I method for the process as a whole. Then, ideas were searched to

innovate process segments and steps. The Table 4 below summarizes a final list of generated ideas.

**Table 4.** Final list of ideas—assembly process.

| Idea/Possible Solution | Resolved Problem/Improved Step |
| --- | --- |
| Re-design of parts to eliminate riveting | Parts are re-designed such that there is no need for riveting. Joints enable parts to be attached just by pushing them together. This leads to a reduction in the assembly time. There is a need to use DFA (Design for Assembly). |
| Assembly at the place of the final installation | By simplifying assembly steps, part of assembly activities can be made at the place of the final installation. This leads to a reduction in process activities at the assembly line, a reduction in the transported volume, and fewer costs. There is a need to use DFA (Design for Assembly). |
| Reorganization of the production—instead of one assembly line, several individual workplaces, i.e., D-Shops (one worker does all activities) | Whole process. This leads to an elimination of transport, an improvement in ergonomy, and better use of workers. |
| Ergonomic improvement in workplaces | Small parts and tools are available in each D-Shop, reducing the time for preparation and assembly. |
| Use of fixtures, jigs, and Poka-Yoke | Faster positioning of parts, easier manipulation, and a reduction in quality checks. |
| During the testing, perform other activities (preparation, etc.) | Time reduction, elimination of waiting time. |
| Computer assistance | Computer, sensors, and signals to assist with mistake-proof assembly. |

It can be seen that several ideas led to changes in the designs of parts in order to eliminate riveting technology. For the purposes of this study, the re-design of parts should not be considered as a solution. A change in product design is a different topic, which is why only ideas focusing on the process will be considered for the purposes of this paper. In reality, a change in parts was also recommended as an excellent opportunity to process innovation. The re-design can be achieved by methods such as the Design for Assembly (DFA), Design for Manufacturing (DFM), or the Design for Six Sigma (DFSS).

The main proposal based on the collected ideas is to reorganize the production. Instead of one assembly line, several one-person assembly stations were proposed. These assembly stations (called D-shops) should be designed so that all or almost all operations on the assembly can be performed in one place by one worker [54]. This is possible because nearly all process activities are based on riveting. As a result, all transportation is eliminated, and more flexibility of the system is achieved. Each D-shop can produce units independently, so there is no need to search for extra workers when someone is missing. Additionally, more than one demand can be produced at the same time. In addition to D-shops, several smaller improvements to assembly activities based on ideas generated by the algorithm were considered, and a new process proposal for the assembly was made.

1. Parts are transported to the gravitation trays by the individual single worker units (D-shops);

2.    A worker takes parts and assembles them by following the assembly manual. During the testing, they use the time for other activities such as preparation or pre-assembly of the next part;

3.    In the second part of the workplace, the assembled unit is packed and transported via a gravitation conveyor to the place for finished parts;

4.    Regular transport takes finished parts and supplies workplaces with parts for assembly.

Each workplace (D-shop) is adjustable to achieve optimal ergonomic conditions for each worker. Additionally, solutions such as a Computer-Assisted Assembly [55], Computer Aided Assembly (CAA) [56], and others are recommended.

## 3. Results

Processes and their improvements described in sections 2.4 and 2.5 were compared. The yield of the first improvement was compared with the yield of improvement from the innovative algorithm. As a measure for comparing the process, the ideality described by Equations (3) and (4) was used. Additionally, different and more common parameters such as production time, number of steps, and others were compared to determine the difference between the classical approach and the use of the algorithm based on TRIZ.

### 3.1. Packing Process (Chandeliers)

The ideality for the process of packing was calculated, firstly, for the original state, where all equation parameters were set to one. See Equation (5).

$$Ip_R = \frac{1+0}{(1+1+1+1+1+1)+(1+1+1+1+0)} = 0.1 \, [-]$$  (5)

Similarly, the ideality was determined for the improved state, where parameters were calculated based on a ratio to the original state. See Table 5.

**Table 5.** Parameters for calculation of process ideality—original to improved state (packing process).

| Parameter | Original State—Real Values | Original State—Equation Coefficients | Improved State—Real Values | Improved State—Equation Coefficients |
|---|---|---|---|---|
| POd | 1 | 1 | 1 | 1 |
| POe | 0 | 0 | 0 | 0 |
| T | 21 | 1 | 17.5 | 0.83 |
| F | 85.5 | 1 | 73.36 | 0.86 |
| P | 8 | 1 | 8 | 1 |
| Eq | 21 | 1 | 21 | 1 |
| E | 8.6 | 1 | 8.6 | 1 |
| M | 299 | 1 | 299 | 1 |
| W | 120 | 1 | 120 | 1 |
| NOK | 0.04 | 1 | 0.04 | 0.96 |
| Sp | 2.06 | 1 | 1.36 | 0.66 |
| Se | 0.8 | 1 | 0.64 | 0.8 |
| O | 0 | 0 | 0 | 0 |

Based on coefficients from the Table 5, a process ideality for the improved state could be calculated; see Equation (6) below.

$$Ip_1 = \frac{1+0}{(0.83+0.86+1+1+1+1)+(1+0.96+0.66+0.8+0)} = 0.101 \left[-\right] \quad (6)$$

The same approach was used for the determination of idealities for the second improvement. Two idealities were calculated for the improved state and the innovated state, where the improved state was set as the reference. See the Table 6.

**Table 6.** Parameters for calculation of process ideality—improved to innovated state (packing process).

| Parameter | Original State—Real Values | Original Stat—Equation Coefficients | Improved State—Real Values | Improved State—Equation Coefficients |
|---|---|---|---|---|
| POd | 1 | 1 | 1 | 1 |
| POe | 0 | 0 | 0 | 0 |
| T | 17.5 | 1 | 8.5 | 0.49 |
| F | 73.36 | 1 | 28.33 | 0.39 |
| P | 8 | 1 | 8 | 1 |
| Eq | 21 | 1 | 2 | 0.1 |
| E | 8.6 | 1 | 6.2 | 0.72 |
| M | 299 | 1 | 12 | 0.04 |
| W | 120 | 1 | 0 | 0 |
| NOK | 0.04 | 1 | 0.02 | 0.51 |
| Sp | 1.36 | 1 | 0.06 | 0.04 |
| Se | 0.64 | 1 | 0.16 | 0.25 |
| O | 0 | 0 | 0 | 0 |

For the second comparison, the improved state was set as the reference state, so all its parameters were set to one in the equation. The parameters for the ideality of the innovated state were calculated from the ratio between the reference state of the improved process. The results are shown in the Table 7 below.

**Table 7.** Comparison of process ideality—packing process.

| Process State | Process Ideality | Ideality Change | Ideality Change |
|---|---|---|---|
| | (-) | (-) | (%) |
| Original (REF.) | 0.1 | - | - |
| Improved (Lean) | 0.101 | 0.001 | 1 |
| Improved (REF.) | 0.1 | - | - |
| Innovated (AP3I) | 0.283 | 0.183 | 183 |

From the table above, a big difference between yields of process ideality from the first improvement and the application of the innovative algorithm is clear. Even though the innovated state was compared with an already improved state, the change in the process ideality is dramatically bigger.

To ensure that the algorithm and process ideality work, different process indicators were also compared; see the Table 8.

**Table 8.** Comparison of improvement by classical indicators—packing process.

| Process State | Average Production Time | VA Index | Number of Steps | Throughput Distance | Walking Distance |
|---|---|---|---|---|---|
| | (min) | (-) | (-) | (m) | (m) |

| | | | | | |
|---|---|---|---|---|---|
| Original | 21.0 | 0.262 | 20 | 82.6 | 78.3 |
| Improved (Lean) | 17.5 | 0.314 | 16 | 47.7 | 59.5 |
| Innovated (AP3I) | 8.5 | 0.412 | 8 | 36.3 | 12.1 |

The average production time represents the average time to pack the part, in other words, the time needed to pack and export the incoming part. VA index is the ratio between the time required by value-adding activities and the production time. It gives information about the amount of value-adding activities in the whole process. The number of process steps is based on the types of activities in the packing process. The throughput distance is the length of the material path through the packing hall. This is one of the original indicators of the process improvement. The walking distance is similar to the throughput distance, but it measures the length of a worker's paths during the packing process. As seen in the table above, even classical indicators show that the proposed process based on the ideas from the innovative algorithm is better than the original or improved state of the packing process. The relative change of the indicators is, in general, made bigger from the application of the algorithm. A comparison of relative improvements can be seen in the Figure 9 below.

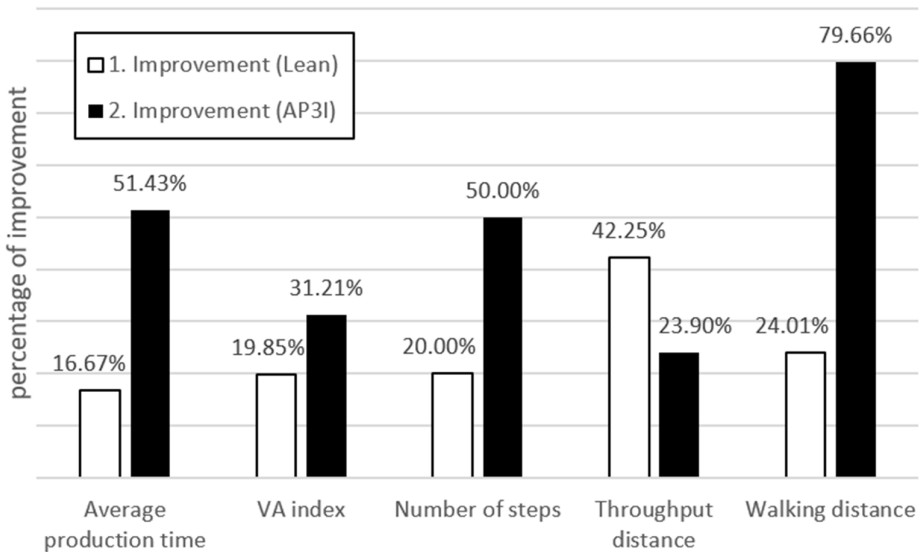

**Figure 9.** Comparison of relative improvement rates of process indicators—packing process.

*3.2. Assembly Line (Heat Exchanger Units)*

The process of assembly of the heat exchanger units was also compared from the point of view of process ideality. The results are summarized in the Table 9 below.

**Table 9.** Comparison of process ideality—assembly line.

| Process State | Process Ideality | Ideality Change | Ideality Change |
|---|---|---|---|
| | (-) | (-) | (%) |
| Original (REF.) | 0.1 | - | - |
| Improved (Lean) | 0.102 | 0.002 | 2 |
| Improved (REF.) | 0.1 | - | - |
| Innovated (AP3I) | 0.11 | 0.01 | 10 |

Innovation in the assembly process, measured by process ideality, is not as dramatic as it is in the case study of the packing. This is probably caused by the choice of the solution. Only the rearrangement of the process was considered, while the re-design of parts was ignored. If parts were re-designed to change the assembly technology, the process ideality would be changed even more.

A new simulation model was created and compared with previous states for the comparison of process indicators. See the Figure 10.

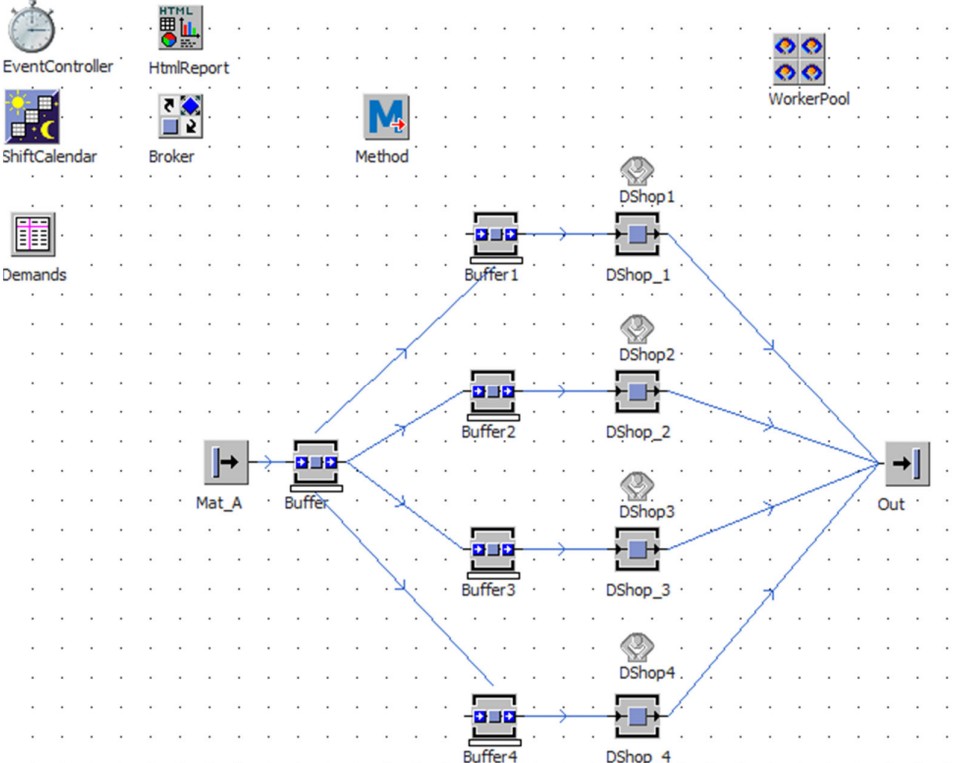

**Figure 10.** Simulation model of the proposed assembly process with D-Shops.

Again, several classical indicators of the process state were used to compare the effectiveness of the improvement due to the simulation and the innovative algorithm. See Table 10.

**Table 10.** Comparison of improvement by classical indicators—assembly line.

| Process State | Average Production Time | VA Index | Number of Steps | Number of Parts | Walking Distance |
|---|---|---|---|---|---|
| | (min) | (-) | (-) | (pcs./10 days) | (m) |
| Original | 40.24 | 0.418 | 40 | 237 | 8.6 |
| Improved (Lean) | 37.89 | 0.444 | 40 | 331 | 8 |
| Innovated (AP3I) | 28.10 | 0.556 | 32 | 401 | 2.9 |

The improvement can be seen from the table above. Even though the algorithm was used on an already improved process, the yield of classical indicators is good. The relative change in indicators is, in general, bigger than from the first improvement. This can be seen in the Figure 11 below.

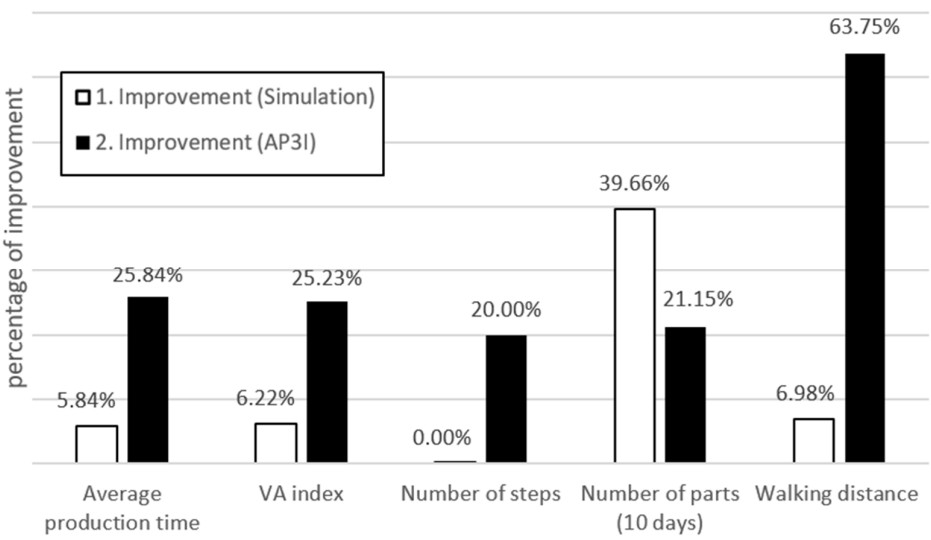

**Figure 11.** The comparison of relative improvement rates of process indicators—assembly line.

*3.3. General Results*

From the results of individual case studies, the average improvement in ideality was determined and compared. See the Figure 12.

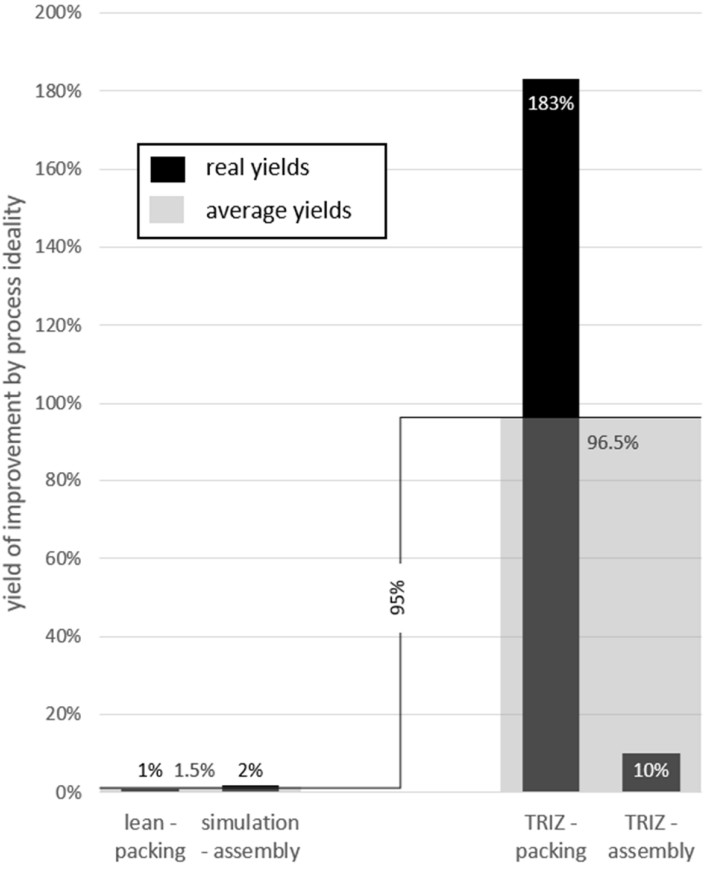

**Figure 12.** Yields of improvement due to the process ideality.

By averaging the yields of improvement in the process ideality from both case studies and comparing them with the average yield of improvement due to AP3I, it can be seen that the difference is 95%.

An overall comparison of classical process indicators can be seen in the Figure 13 below.

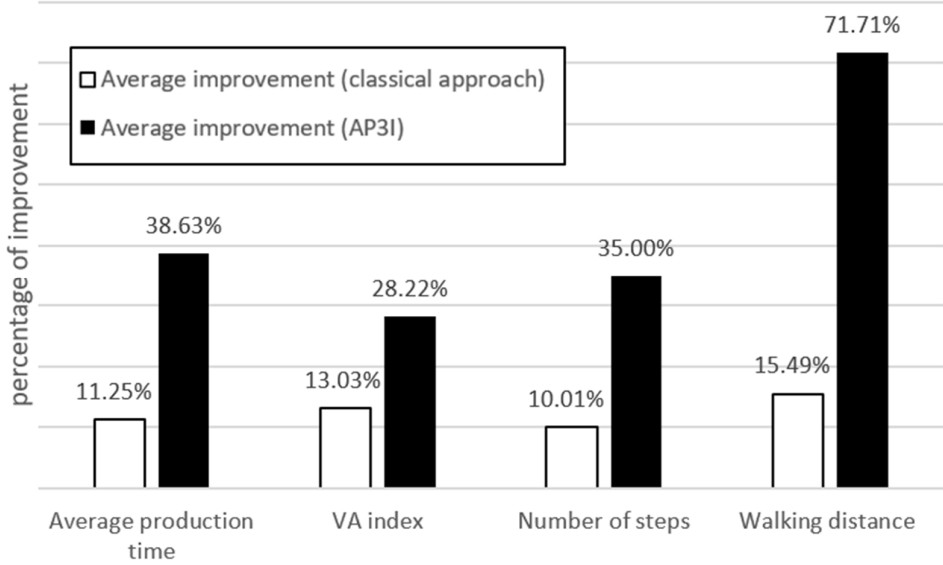

**Figure 13.** Overall relative improvement rates of process indicators.

The improvement rates of process indicators from both case studies were averaged and compared. This comparison provides a more general result than a separate comparison for individual processes. More comparisons with many improved processes should be made for future work to determine the general improvement rate trend.

## 4. Discussion

As shown by the results of the presented case studies, the use of the algorithm can bring more radical ideas for process improvement. Compared with classic methods, which mainly focus on optimizing the process, which could be its limitation, the use of the proposed algorithm can bring a bigger improvement rate. The proposed algorithm (AP3I) and TRIZ are tools that focus on the radical innovation of the process. The improvement is based on the change in the process principle, by which fewer process steps or resources are needed, while the same or better result is achieved.

The average improvement measured by process ideality is 95%, which can be marked as a radical change compared to the average improvement in initial projects of 1.5%. In the study with the assembly line, even better results would be achieved if the re-design of parts is considered. Even classical process indicators positively changed more than double using the AP3I algorithm compared to the change from the first improvement.

A limitation is the small number of applications of the algorithm so far. The presented studies have given only a rough estimation of what may be achieved using the algorithm. The real verified values can be found after more applications to many processes. This is also a proposal for the next work. The proposed algorithm should soon be applied to many different processes to obtain deeper functionality verification and understand the possible improvement rates.

Based on several studies in which Lean, Six Sigma, or Lean Six Sigma methods were reviewed or used to improve processes, the improvement rate ranges primarily from units of 1% to 30%. There are many cases where the improvement rate is much more significant,

but it seems that these cases are not as often. These findings are based on the research of other authors such as [57–68]. Based on a comparison of the classical indicators, the change achieved in the process is slightly above average. Because the principle of the process was just changed, there is a high probability that by using the classical methods such as Lean or Six Sigma to optimize the new process state, the improvement will be more significant. This premise is based on the S-curve development of technical systems, shown in the Figure 14 below.

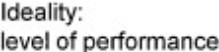

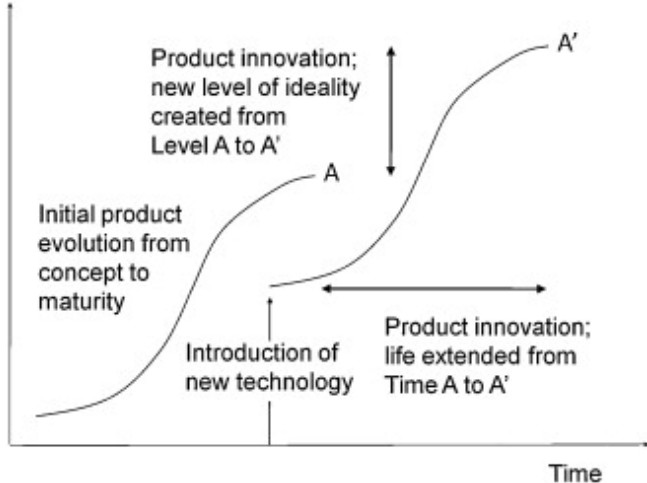

**Figure 14.** Evolutionary S-curve, showing technology shift effect [69].

It is clearly shown that if a new principle is found, some parameters or functionality could worsen compared to the previous principle (A). However, the new principle can evolve into a more ideal system that fulfills the purpose even more efficiently, with fewer resources, fewer harms, and more outcomes (A').

## 5. Conclusions

The algorithm for Process Innovation by Increasing Ideality was developed based on the review. For verification purposes, a metric of the process ideality was redefined in order to be clear and possible to use. The algorithm was tested on two real processes. Firstly was the packing process of glass and metal parts of chandeliers. This process was initially improved by Lean principles, where material flows were optimized and the layout of the whole packing hall was re-designed. Additionally, several smaller improvements were applied. The algorithm was used for the improved process. Proposals from the algorithm change the concept of the packing principle from boxes with different packing protective materials to frames with flexible membranes. From the point of view of process ideality, ideality changed initially by 1% and after using the algorithm by 183%. The second process was an assembly line for heat exchanger units. This process was firstly improved by the use of a computer simulation and material flow optimization, where the focus was mainly on productivity. The line was slightly re-designed, and several buffers were added. Due to the application of the algorithm, a new process was proposed. Instead of one big production line, several small one-person workshops (D-Shops) were proposed. Because most operations are riveting, there is no need for much space. Therefore, the process is changed to make the process more dynamic. Additionally, a proposal for a change in parts in order to reduce riveting was set. For this study, this proposal was ignored because it is a discipline of product innovation. In reality, combined improvement would be

much more significant. From the point of view of process ideality, the first improvement was by 2% and the improvement from the second proposal was by 10%,

Processes were also compared using classical indicators such as production time, number of steps, VA index, and walking distance. Most of these parameters achieved a bigger improvement rate from the improvement due to the proposed algorithm than from the initial improvement.

The algorithm needs to be used on more processes to obtain more data about possible improvement rates. However, even after these two applications, it is clear that the algorithm has a big potential to be a powerful tool for radical process innovation. Additionally, it can complete the toolset of industrial engineers, where there is no powerful tool for radical process innovation.

**Author Contributions:** Conceptualization, V.S. and P.L.; methodology, V.S.; validation, V.S.; writing—original draft preparation, V.S.; writing—review and editing, P.L.; visualization, V.S.; supervision, P.L.; project administration, V.S. All authors have read and agreed to the published version of the manuscript.

**Funding:** This work was supported by the Student Grant Competition of the Technical University of Liberec under the project No. SGS-2020-5027—Research of new approaches to process improvement.

**Institutional Review Board Statement:** Not applicable.

**Informed Consent Statement:** Not applicable.

**Conflicts of Interest:** The authors declare no conflicts of interest.

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
