# Peer review of "Algorithm for Process Innovation by Increasing Ideality"

_processes, doi:10.3390/pr10071283_

Round 1

Reviewer 1 Report

11.      It is worth avoiding abbreviations and abbreviations in the summary and keywords

22.      Why was there introduced subsection 1.1 in Introduction if there is no 1.2 and beyond?

33.      The authors abuse the use of abbreviations and abbreviations. Why introduce them if they are not used in the article after they are introduced. Example  -  Just-in-time (JIT) – line 63, Lean/Six Sigma (LSS) – line 77 and others

44.      It is enough to introduce abbreviation once. You don't have to do this several times, for example (3I) – line 210, 224

55.      Figures 7-8 are poorly legible with small type

66.      The references list does not meet the journal's formatting requirements. See template

77.      There is no requirement to include an ISBN for books in your references list

88.      You must supply DOI for all references that have them. Example – Nr 6 (line 592), DOI - 10.1109/RAMS.2015.7105136 and others

99.      You must provide a DOI for all articles or an access link. Example – Nr 7 (line 594), Nr 11 (line 603) and others

110.  The same letters in formulas are often described differently, which is very confusing. Example – letter B (line 123, 131, 134, 137 …). It's best to clear it up only after the first use and don't repeat it anymore.

111.  In the sums of the indicators (åB, åH …), it is not entirely clear whether they are limited in any number or may not be finished? Maybe I would be better to give in this look: åBi, åHj, where i - ...., j - …

Author Response

Dear Reviewer 1,
Thank you for the suggestions and the opportunity to improve our paper.

Please see the attachment for the full response.

Thank you,
Authors.

Reviewer 2 Report

Original research manuscript entitled "Algorithm for Process Innovation by Increasing Ideality" of interest for a journal like "Processes". Presented study opens up wide prospects in this field of knowledge. Overall, the topic of this study is relevant, and the manuscript was well organized and written.

In my opinion, methodological inaccuracies are not detected. This work introduces a new comprehensive algorithm for the innovation of processes in production based on TRIZ principles. The Algorithm for Process Innovation by Increasing Ideality (AP3I) helps search for innovative ways to improve or change the process – whole or its segments. The algorithm was tested on two real processes.

This work includes the next harmonious structure: Introduction (p. 1 – 2); Methods (p. 3 – 14); Results (p. 14 – 20); Discussion (p. 20 – 21); Conclusions (p. 21 – 22). The figures and tables are appropriate, they properly show the data. So, they easy to interpret and understand by readers.

Overall, this work is ready for publication in present form.

Author Response

Dear Reviewer 2,

We thank you for your positive review of our manuscript.

With regards,
Authors

Round 2

Reviewer 1 Report

 Accept in present form